# Intestinal parasitic infections and associated factors among street dwellers and prison inmates: A systematic review and meta-analysis

Daniel Getacher Feleke[1]*, Yonas Alemu[1], Habtye Bisetegn[2], Melat Mekonnen[2], Nebiyou Yemanebrhane[3]

1 Department of Microbiology, Immunology and Parasitology, College of Health Sciences, Addis Ababa University, Addis Ababa, Ethiopia, 2 Department of Medical laboratory Science, College of Medicine and Health Sciences, Wollo University, Dessie, Ethiopia, 3 Ethiopian Public Health Institute, Addis Ababa, Ethiopia

* danibest2002@gmail.com

**Data Availability Statement:** All relevant data are within the manuscript and its Supporting information files.

## Abstract

### Background

Intestinal parasitic infections are closely associated with low household income, poor personal and environmental sanitation, and overcrowding, limited access to clean water, tropical climate and low altitude. Street dwellers and prisoners are forced to live in deprived situations characterized by inadequate facilities. Therefore, this study aimed to estimate the pooled prevalence and associated factors of intestinal parasitic infections among street dwellers and prison inmates.

### Method

Study searches were carried out in Electronic data bases such as PubMed/Medline, HINARI, EMBASE, Science Direct, Scopus, Google Scholar and Cochrane Library. Studies published only in English and have high quality Newcastle Ottawa Scale (NOS) scores were included for analysis using Stata version 14 software. Random-effects meta-analysis model was used for analysis. Heterogeneity was assessed using the Cochrane's Q test and $I^2$ test statistics with its corresponding p-values. Moreover, subgroup, sensitivity analyses and publication bias were computed.

### Result

Seventeen eligible studies consist of 4,544 study participants were included. Majority of the study participants were males (83.5%) and the mean age of the study participants was 25.7 years old. The pooled prevalence of intestinal parasitic infections among street dwellers and prison inmates was 43.68% (95% CI 30.56, 56.79). Sub-group analysis showed that the overall pooled prevalence of intestinal parasitic infections among prison inmates and street dwellers was 30.12% (95%CI: 19.61, 40.62) and 68.39% (95%CI: 57.30, 79.49),

**Funding:** The authors received no specific funding for this work.

**Competing interests:** The authors have declared that no competing interests exist.

respectively. There was statistically significant association between untrimmed fingernail and intestinal parasitic infections (AOR: 1.09 (95%CI: 0.53, 2.23).

## Conclusion

In this study, the pooled prevalence of intestinal parasitic infections among street dwellers and prison inmates was relatively high. Fingernail status had statistically significant association with intestinal parasitic infection. The prevention and control strategy of intestinal parasitic infection should also target socially deprived segment of the population such as street dwellers and prison inmates.

## Introduction

Intestinal parasitic infections (IPI) are caused by intestinal helminths and protozoan parasites which still pose one of the major public health problems in developing countries where adequate water and sanitation facilities are lacking [1, 2]. Globally, about 3.5 billion people were infected, and of that 450 million are ill as a result of one or more intestinal parasitic infections [3]. It is estimated that more than 10.5 million new cases are reported annually and *Trichuris trichiura*, hookworms, *Ascaris lumbricoides*, *Schistosoma* species, *Giardia lamblia* and *Entamoeba histolytica* are the most common intestinal parasites [4].

These infections may lead to malnutrition, malabsorption, anemia, intestinal obstruction, mental and physical growth retardation, diarrhea, impaired work capacity, and reduced growth rate constituting important health and social problems [5, 6]. Intestinal parasitic infections are more prevalent among the poor segment of the population and closely associated with low household income, poor personal and environmental sanitation, and overcrowding, limited access to clean water, tropical climate and low altitude [7, 8].

Street dwellers are among the most deprived people in urban areas as they are highly affected by low socioeconomic conditions, poor personal and environmental hygiene, and have limited access to clean water [9]. Being street dweller is an increasing problem worldwide with an approximately about 500 million people in the world are homeless [10, 11]. Similarly, prison inmates are among the vulnerable groups to intestinal parasitic infections. In developing countries, prison inmates live in deprived situations characterized by inadequate facilities, malnutrition, scarce potable water, over-crowdedness, and poor hygiene [12]. Besides this, prisoners have no control of their environment in which they live, which puts them at risk of infection with intestinal parasites [13]. Limited healthcare, high risk behaviors, lower immunity due to stress and poor nutrition adds the risks [14].

Street dwellers and prison inmates represent the marginalized communities and the overall low levels of living standards make them prone to parasitic illness than the general population [15]. The prevalence of intestinal parasitic infection among prison inmates showed 42.6%–72.73% in Ethiopia [16–19], 24.7% in Kenya [20], 14.4% and 22.8% in Nigeria [13, 21], 6% in Nepal [22], 7.89% in India [23], 26.5% in Malaysia [14] and 20.2% in Brazil [24]. Whereas, the prevalence of intestinal parasites among street dwellers was reported as 43.9%–89.7% in Ethiopia [25–28], 66.3% in Peru [29] and 71.7% in Sudan [30].

Despite different single studies reporting the prevalence of IPIs and its associated factors among prison inmates and street dwellers, there is no study that systematically compiled IPIs burden among these groups. It would be highly relevant for policy makers and program planners to include them in implementing efficient interventions to decrease the burden and

impacts of IPIs. Therefore, this systematic review and meta-analysis aimed to estimate the pooled prevalence of IPIs and associated factors among street dwellers and prison inmates globally.

## Methods

### Study protocol and registration

The protocol of this systematic review and meta- analysis was registered (CRD42021229664) on the International Prospective Register of Systematic Reviews (PROSPERO)database and the result was reported based on the Preferred Reporting Items for Systematic Review and Meta-analysis' (PRISMA) guidelines [31].

### Search strategy

Electronic data bases such as PubMed/Medline, HINARI, EMBASE, Science Direct, Scopus, Google Scholar, Cochrane Library and thesis and other documentations of universities were thoroughly searched for studies conducted on the topic related to the present systematic review and meta-analysis. Furthermore, manual search for identifying any relevant studies were done whenever necessary. Studies conducted on the prevalence and associated risk factors of intestinal parasitic infections among street dwellers and prison inmates were included in this meta-analysis. The key words used during search were "prevalence,"magnitude","burden","intestinal parasitic infection, "opportunistic intestinal parasitic infections, "associated factors, "contributing factors, "risk factors, "prisoner, "inmates", "prison inmates", "Jail", street dwellers, street living people, homeless and "beggars". These key words were used to search separately or in combination using the "AND"and"OR"Boolean operators. Moreover, reference lists of the included studies were searched for any additional relevant articles.

### Study selection criteria

In this systematic review and meta-analysis, observational studies reported the magnitude and associated risk factors of intestinal parasitic infections among street dwellers and prison inmates worldwide from 1$^{st}$January 2000 to 1$^{st}$December 2020 were considered. With regard to language, only studies reported in English language were included. Case reports, qualitative studies, trials, reviews, letters, conference proceedings, news, studies that did not report the prevalence and associated factors of intestinal parasitic infections among street dwellers and prison inmates were excluded from this study. Whenever the full text of studies is not available, the corresponding authors of the article were communicated for full texts. Abstracts were excluded when the authors didn't respond to our request for the full texts.

### Outcome of measurement

In this study, the pooled prevalence of intestinal parasitic infections and risk factors of intestinal parasitic infections among street dwellers and prison inmates were the outcome of measurement. Odds ratio was used for assessing the association between intestinal parasitic infections and associated factors among street dwellers and prison inmates.

### Data extraction

Systematically searched studies from databases were imported to Endnote version 7 and duplicated studies were removed. Data extraction format was adapted from the Joanna Briggs Institute (JBI) data extraction format [32]. Then the format was prepared on Microsoft Excel sheets by authors (DGF, HB, YA, MM and NY) based on the objective of this systematic review and

meta-analysis. The titles and abstracts of each study were screened by authors independently. All authors (DGF, HB, YA, MM and NY) assessed the full texts of studies based on the pre-specified selection criteria. When disagreements occur among authors during assessment it was solved by discussion. Data from the eligible studies were extracted using standardized data extraction format. The data extraction format includes information about name of the first author, publication year, the country of the study conducted, sample size, study design, prevalence of intestinal parasitic infections, associated factors for intestinal parasitic infections and quality of each study.

## Quality assessment

The quality of included studies was assessed using the Newcastle Ottawa Scale (NOS) for cross-sectional study quality assessment tool [33]. Newcastle Ottawa Scale (NOS) was used to assess methodological quality, comparability, and outcome of each study by all authors (DGF, MM, HB, YA and NY) independently. The cut off point for good quality study was 7 out of 10 which was declared based on previous relevant studies [34]. In this study, all articles were included because they scored more than seven and above in the NOS quality assessment criteria.

## Data processing and analysis

Data were imported to STATA version14 software (Stata Corporation, College Station, Texas 77845 USA) for analysis. The commands used in Stata were based on previous resource [35]. Random-effects meta-analysis model was used for analysis. DerSimonian–Laird method was used to estimate the between-study variance. The Cochrane's Q test (Chi-square) and $I^2$ (%) with its corresponding p-values was used to assess the heterogeneity of the included studies [36, 37]. Subgroup analysis was also computed to investigate the possible source of heterogeneity. Furthermore, sensitivity analyses were done to observe whether the step by step omission of a single study from the analyses influenced the overall pooled prevalence of intestinal parasitic infections. Publication bias was assessed using symmetry of funnel plot and Egger's test statistics. The pooled prevalence of intestinal parasitic infections was reported with a 95%CI and P-values<0.05 were considered statistically significant. Furthermore, the strength of association between intestinal parasites and associated factors was determined using odds ratio.

## Results

### Study selection

Systematic search of studies on the prevalence and associated factors of intestinal parasitic infections among street dwellers and prison inmates identified 123 records. After regress screening of these studies for duplication and eligibility, 17 studies were found eligible and included in this systematic review and meta-analysis (Fig 1).

### Characteristics of included studies

This systematic review and meta- analysis included 17 eligible cross sectional studies consists of 4,544 study participants. Among the included studies eleven of them were done on prison inmates that contain a total of 3,050 study participants. Majority of the study participants were males (83.5%). The mean age of the study participants was 25.7 years old. The sample size of the included studies varied from 114 [23] to 510 [24]. With regard to the prevalence of intestinal parasitic infections, the lowest prevalence (6%) was reported from a study conducted in

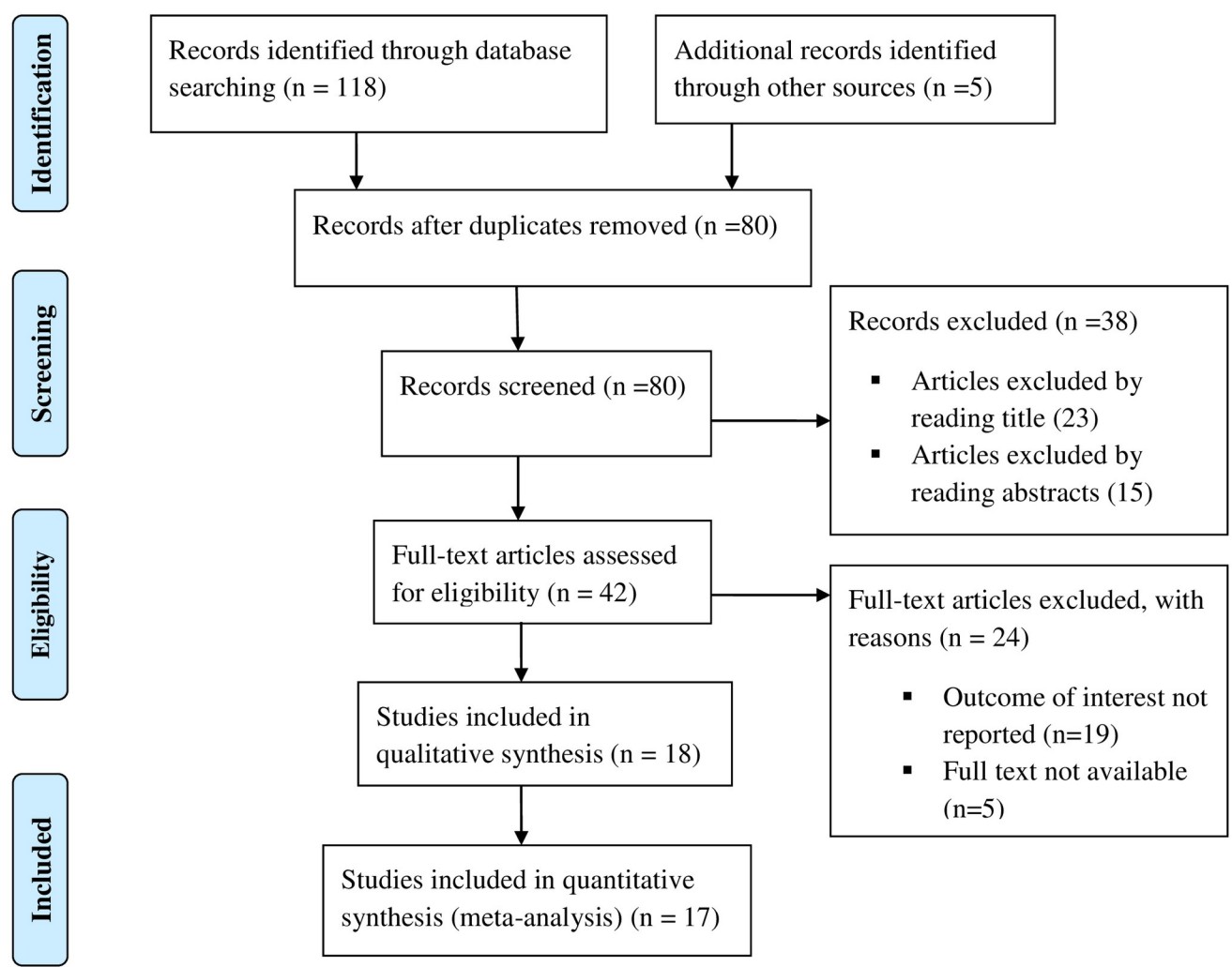

**Fig 1. Flow chart of study selection for systematic review and meta-analysis on the prevalence and associated factors of intestinal parasitic infections among street living people and prison inmates.**

Nepal, Kathmandu prison [22] and the highest (89.7%) was observed from a study done among street beggars in Jimma, Ethiopia [27].

All the included studies were reported from Africa, Asia and Latin America continents. Eight of the studies, 4 among street dwellers [25–28] and another 4 among prison inmates [16–19] were conducted in Ethiopia. Two studies conducted among prison inmates were reported from Nigeria [13, 21]. Malaysia, Nepal, Brazil, Kenya and India reported one study each on prison inmates [14, 20, 22–24]. The remaining two studies on street dwellers were done in Sudan and Peru [29, 30] (Table 1).

## Intestinal parasitic infection prevalence among street dwellers and prison inmates

The pooled prevalence of intestinal parasitic infections among street dwellers and prison inmates was 43.68% (95% CI 30.56, 56.79). DerSimonian-Laired random effects model was used for analysis due to the considerably high heterogeneity between the included studies ($I^2$ = 99.20, p<0.00) (Fig 2).

**Table 1. Characteristics of included studies in the systematic review and meta-analysis of intestinal parasitic infection among street living people and prison inmates.**

| No | Author/ref | Publication year | Country | Study group | Sample size | Number of Cases | Prevalence % | Study quality |
|----|-----------|-----------------|---------|-------------|-------------|-----------------|--------------|---------------|
| 1 | Ameya et al | 2019 | Ethiopia | Prison inmates | 320 | 154 | 48.1 | 9 |
| 2 | Angal et al | 2015 | Malaysia | Prison inmates | 294 | 78 | 26.5 | 9 |
| 3 | Mardu et al | 2017 | Ethiopia | Prison inmates | 291 | 124 | 42.6 | 9 |
| 4 | Terefe et al | 2015 | Ethiopia | Prison inmates | 234 | 111 | 47.4 | 8 |
| 5 | Shrestha et al | 2019 | Nepal | Prison inmates | 400 | 24 | 6.0 | 8 |
| 6 | Amit et al | 2016 | India | Prison inmates | 114 | 9 | 7.9 | 7 |
| 7 | Ahmed et al | 2016 | Nigeria | Prison inmates | 132 | 19 | 14.4 | 7 |
| 8 | Nadabo et al | 2019 | Nigeria | Prison inmates | 250 | 57 | 22.8 | 7 |
| 9 | Mamo | 2014 | Ethiopia | Prison inmates | 121 | 88 | 72.7 | 9 |
| 10 | Curval et al | 2017 | Brazil | Prison inmates | 510 | 103 | 20.2 | 9 |
| 11 | Rob et al | 2016 | Kenya | Prison inmates | 384 | 95 | 24.7 | 9 |
| 12 | Feleke et al | 2019 | Ethiopia | Street dwellers | 246 | 108 | 43.9 | 9 |
| 13 | Bailey et al | 2013 | Peru | Street dwellers | 258 | 171 | 66.3 | 7 |
| 14 | Mekonnen et al | 2014 | Ethiopia | Street dwellers | 355 | 255 | 71.8 | 8 |
| 15 | Kheir et al | 2017 | Sudan | Street dwellers | 207 | 148 | 71.7 | 7 |
| 16 | Zenu et al | 2019 | Ethiopia | Street dwellers | 312 | 208 | 66.7 | 9 |
| 17 | Lakew et al | 2015 | Ethiopia | Street dwellers | 116 | 104 | 89.7 | 7 |

## Sub-group analysis

In this systematic review and meta-analysis, the heterogeneity of included studies was considerable ($I^2$ = 99.2%, p<0.00). Therefore, sub-group analysis was performed based on study group, sample size, publication year and continent to investigate the source of heterogeneity. In prison inmates, eleven studies that contain 3,050 individuals were included and analyzed. The overall pooled prevalence of intestinal parasitic infections was 30.12% (95%CI: 19.61, 40.62) and the heterogeneity was considerably high ($I^2$ = 98.3, p<0.00) (Fig 3).

Analysis of six studies comprised 1,494 study participants among street dwellers were analyzed and showed an overall pooled intestinal parasitic infection prevalence of 68.39% (95%CI: 57.30, 79.49) ($I^2$ = 95.8%, p<0.00) (Fig 4).

Furthermore, studies with sample size ≥300 had higher overall pooled prevalence of intestinal parasitic infections (47.68% (32.50, 62.86)) than studies with sample size >300. Moreover, sub-group analysis based on publication year showed that studies published from 2000 to 2015 had higher pooled intestinal parasitic infection prevalence (62.38% (43.38, 81.38)) and the African continent had the highest pooled prevalence of intestinal parasitic infections 51.35% (37.78, 64.92) compared to the Asia and Latin America sub-group (Table 2).

## Prevalence of some common intestinal parasites among street dwellers and prison inmates

In this study, the prevalence of common species of intestinal parasitic infections among street dwellers and prison inmates was pooled. Among the reported intestinal parasite species *A. lumbricoides* was the dominant species with an overall pooled prevalence of 14.07% (95% CI: 9.15, 18.99). On the other hand, *E. histolytica* and *G. lamblia* were the two dominant intestinal protozoa with an overall pooled prevalence of 7.83% (5.47, 10.18) and 7.49% (5.08, 9.90), respectively. The least common parasite species reported was *E. vermicularis* 1.69 (0.20, 3.18) (Table 3).

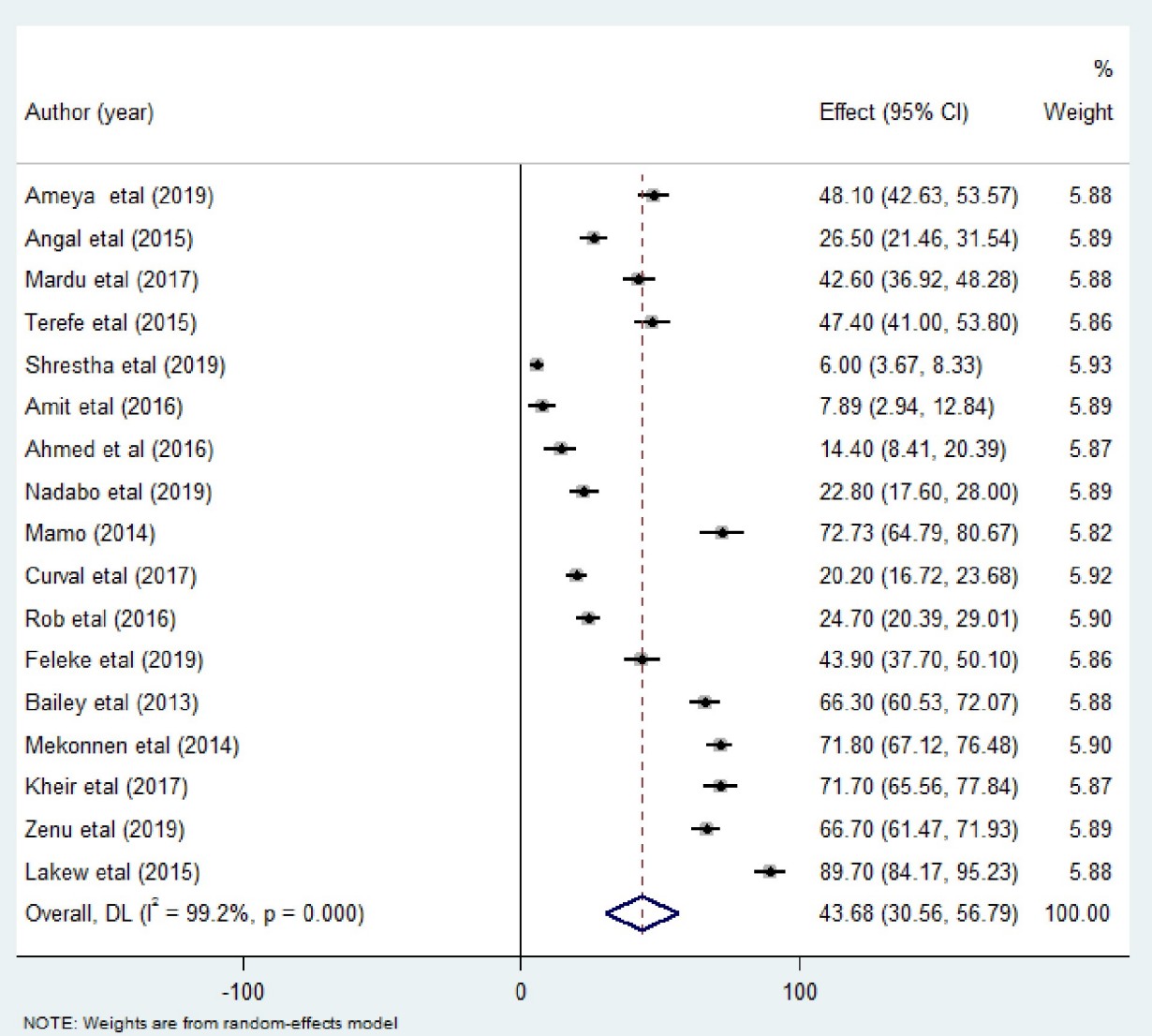

**Fig 2. Forest plot showing the pooled prevalence estimate of intestinal parasitic infection among street dwellers and prison inmates.**

## Publication bias

The funnel plot showed the distribution of the included studies was asymmetrical (Fig 5).

## Sensitivity analysis

Sensitivity analysis was performed to observe whether the overall pooled intestinal parasite prevalence significantly changed when each included study omitted step by step from the analysis. The sensitivity analysis showed that none of the included studies significantly altered the combined estimates of intestinal parasitic infections (Table 4).

## Factors associated with intestinal parasitic infections among street dwellers and prison inmates

In this meta-analysis, factors such as hand washing habit, residence and fingernail status were analyzed. Four studies [18, 24, 26, 28] were assessed for the association between hand washing

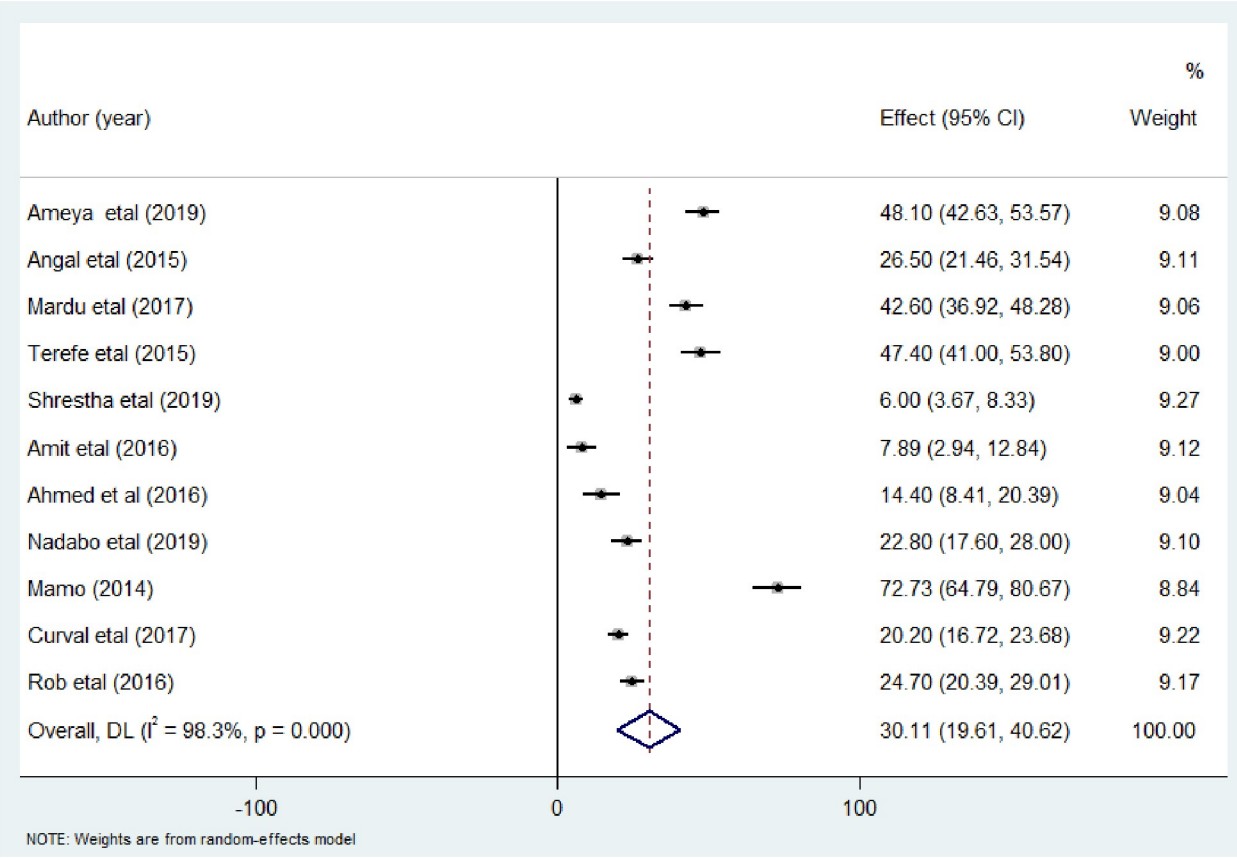

**Fig 3. Forest plot showing the pooled prevalence of intestinal parasitic infection among prison inmates.**

habit and the prevalence of intestinal parasitic infections among street dwellers and prison inmates. Although there was no statistically significant association between intestinal parasitic infection prevalence and hand washing habit, the prevalence of intestinal parasitic infections among street dwellers and prison inmates who didn't have hand washing habit was 2.73 times higher than those who had hand washing habit (AOR: 2.73, (95% CI: 1.83, 4.07)). There was low heterogeneity within the studies ($I^2 = 0.0\%$ and P = 0.726) (Fig 6).

The association between residence and intestinal parasitic infection was analyzed in four studies [16–18, 22]. All the four included studies were conducted among prison inmates. The analysis showed that rural resident were less likely to be infected by intestinal parasitic infection compared to urban residents (AOR: 0.67, (95%CI: 0.33, 1.34). Random-effects meta-analysis was used due to the considerable heterogeneity ($I^2 = 82.7\%$; P = 0.001) within the studies (Fig 7).

Finally, the association between fingernail status and intestinal parasitic infections among street dwellers and prison inmates has been computed in five included studies [16, 17, 25, 26, 28]. There was statistically significant association between untrimmed fingernail and intestinal parasitic infections as compared to trimmed fingernail (AOR: 1.09 (95%CI: 0.53, 2.23)). There was evidence of high heterogeneity across the included studies ($I^2 = 78.0\%$; P = 0.001) (Fig 8).

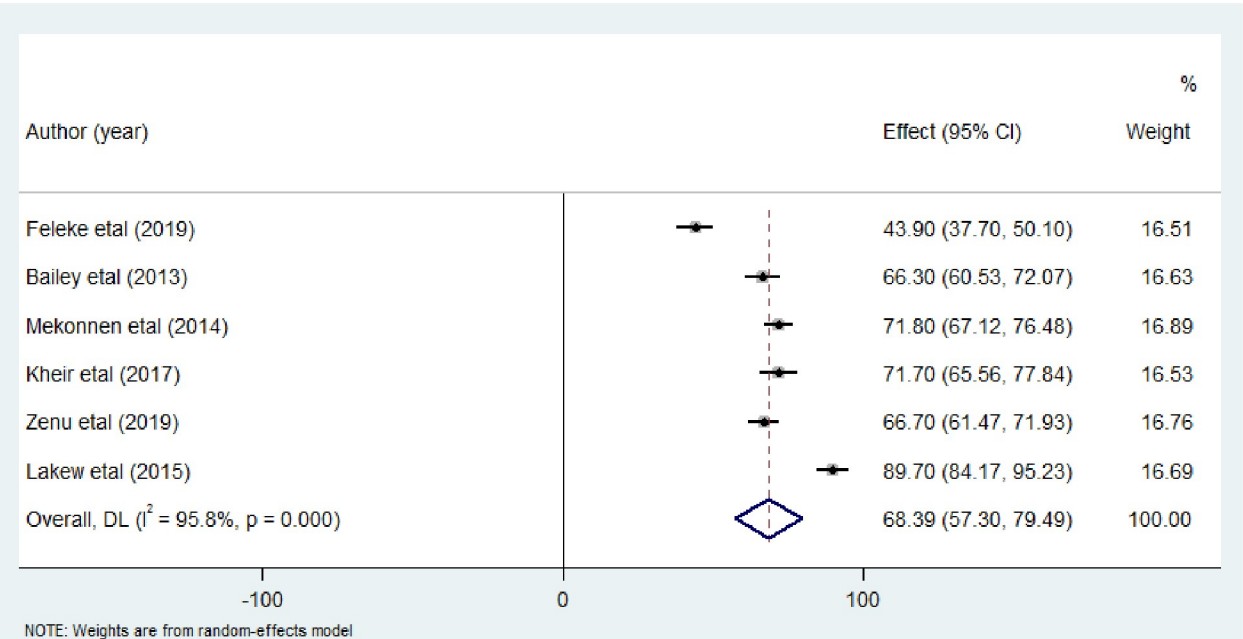

**Fig 4. Forest plot showing the pooled prevalence estimate of intestinal parasitic infection among street dwellers.**

**Table 2. The pooled prevalence of intestinal parasitic infection among street dwellers and prison inmates based on subgroup analysis.**

| Variables | Sub-groups | Included studies | Sample size | Prevalence% (95% CI) | $I^2$ & P-value |
|---|---|---|---|---|---|
| Sample size | ≥300 | 12 | 2575 | 47.68 (32.50, 62.86) | 98.8%, 0.00 |
| | >300 | 5 | 1969 | 34.10 (11.72, 56.48) | 99.4%, 0.00 |
| Continent | Africa | 12 | 2968 | 51.35 (37.78, 64.92) | 98.6%, 0.00 |
| | Asia, Latin America | 5 | 1576 | 25.28 (7.74, 42.82) | 99.0%, 0.00 |
| Publication Year | 2000–2015 | 6 | 1378 | 62.38 (43.38, 81.38) | 98.5%, 0.00 |
| | 2016–2021 | 11 | 3166 | 33.48 (20.33, 46.63) | 98.9%, 0.00 |

**Table 3. Pooled prevalence of intestinal parasitic infection among street dwellers and prison inmates by intestinal parasites species.**

| Type of intestinal parasite species | Pooled prevalence 95% CI | $I^2$ & P-value |
|---|---|---|
| A.lumbricoides | 14.07% (9.15, 18.99) | 98.3, P = 0.00 |
| E.histolytica/dispar | 7.83% (5.47, 10.18) | 94.1%, P = 0.00 |
| G. lamblia | 7.49% (5.08, 9.90) | 95.8%, P = 0.00 |
| T.trichuria | 6.59 (4.29, 8.90) | 96.2%, P = 0.00 |
| H.nana | 5.49 (2.61, 8.37) | 84.4%, P = 0.00 |
| S.mansoni | 4.71 (0.94, 8.49) | 90.0%, P = 0.00 |
| Taenia spp. | 3.78 (1.76, 5.80) | 93.4%, P = 0.00 |
| S.sterocoralis | 2.81 (1.35, 4.28) | 84.9%, P = 0.00 |
| Hookworm | 2.70 (1.60, 3.81) | 83.1%, P = 0.00 |
| E.vermicularis | 1.69 (0.20, 3.18) | 49.6%, P = 0.137 |

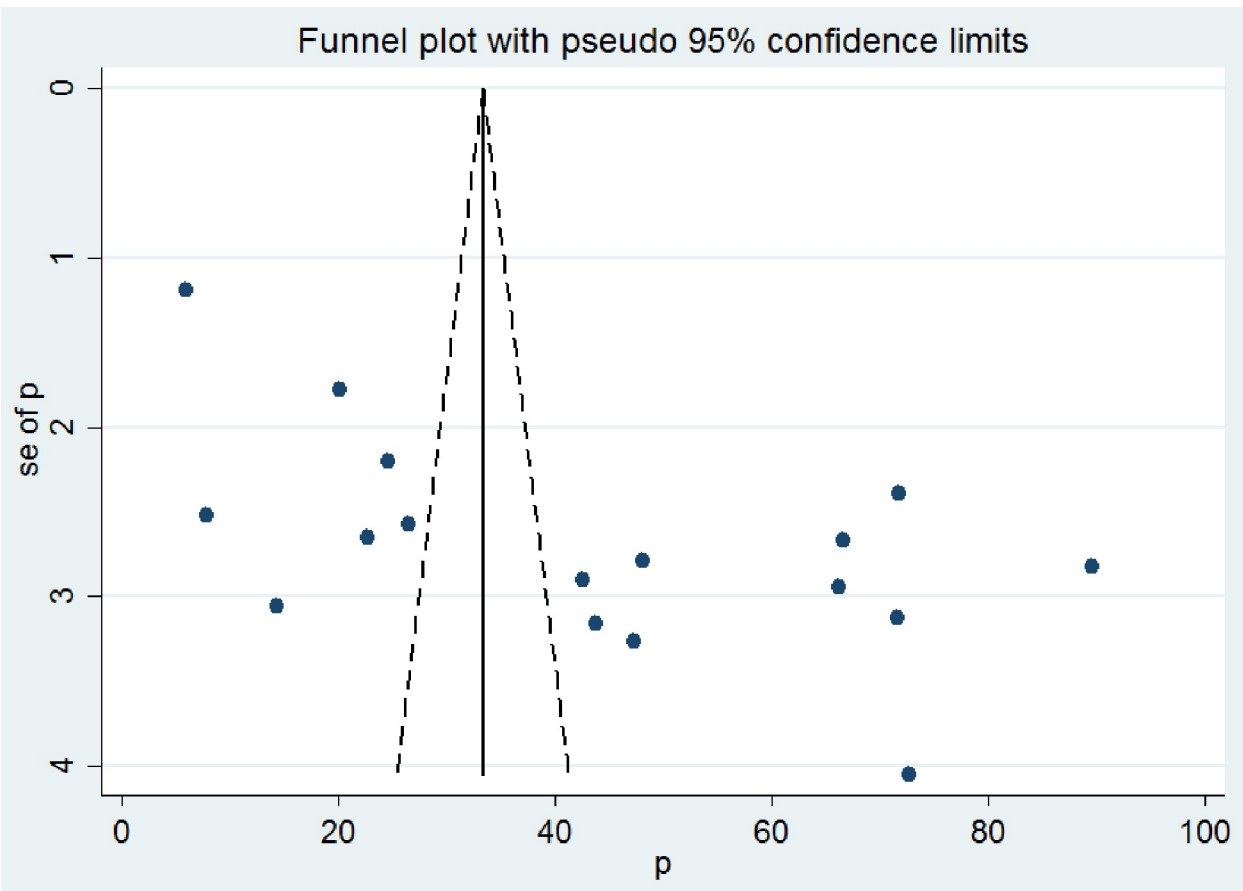

**Fig 5. Funnel plots showing publication bias test of the included studies in this meta-analysis.**

**Table 4. Sensitivity analysis of intestinal parasitic infections among street dwellers and prison inmates.**

| Omitted studies | Pooled prevalence (%) 95% CI |
|---|---|
| Ameya et al | 43.40 (29.63, 57.17) |
| Angal et al | 44.75 (30.84, 58.66) |
| Mardu et al | 43.75 (29.93, 57.56) |
| Terefe et al | 43.45 (29.73, 57.16) |
| Shrestha et al | 46.04 (33.71, 58.38) |
| Amit et al | 45.92 (32.34, 59.49) |
| Ahmed et al | 45.50 (31.83, 59.18) |
| Nadabo et al | 44.98 (31.13, 58.84) |
| Mamo | 41.88 (28.53, 55.23) |
| Curval et al | 45.16 (30.99, 59.32) |
| Rob et al | 44.87 (30.84, 58.90) |
| Feleke et al | 43.66 (29.91, 57.42) |
| Bailey et al | 42.26 (28.88, 55.65) |
| Mekonnen et al | 41.91 (28.89, 54.92) |
| Kheir et al | 43.68 (30.57, 56.79) |
| Zenu et al | 42.23 (28.90, 55.57) |
| Lakew et al | 39.05 (26.70, 51.41) |
| **Overall** | **43.68 (30.56, 56.79)** |

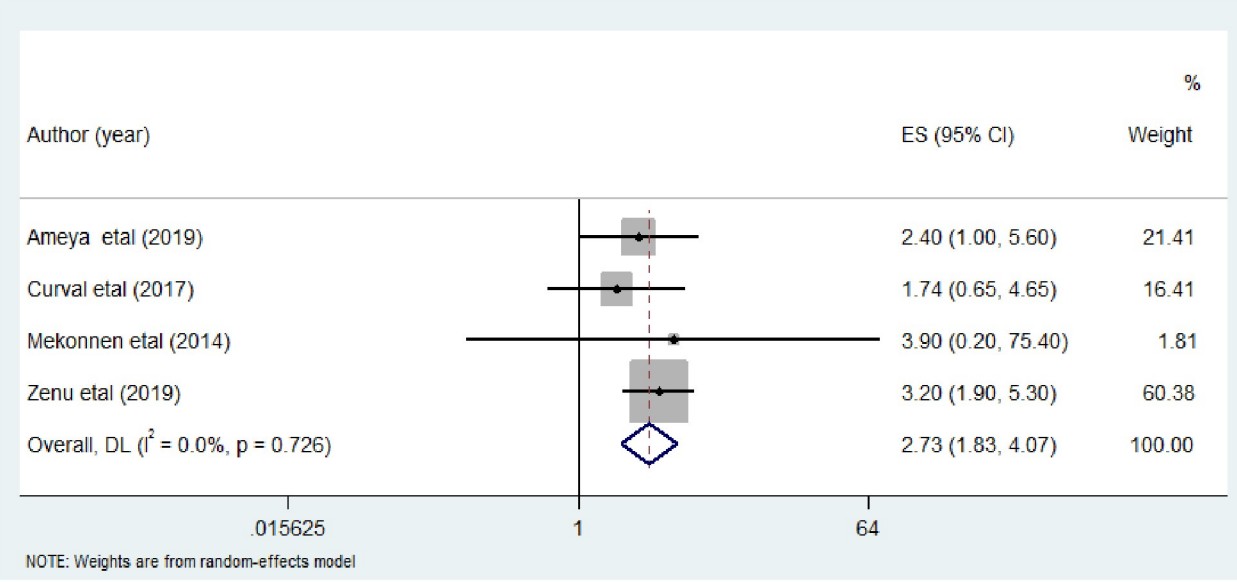

**Fig 6. The pooled odds ratio of the association between hand washing habit and intestinal parasitic infection among street dwellers and prison inmates.**

## Discussions

Intestinal parasitic infections are still one of the major public health problems in developing countries where adequate water supply and sanitation facilities are scarce [1, 2]. This systematic review and meta-analysis aimed to assess the pooled prevalence and associated factors of intestinal parasitic infections among street dwellers and prison inmates.

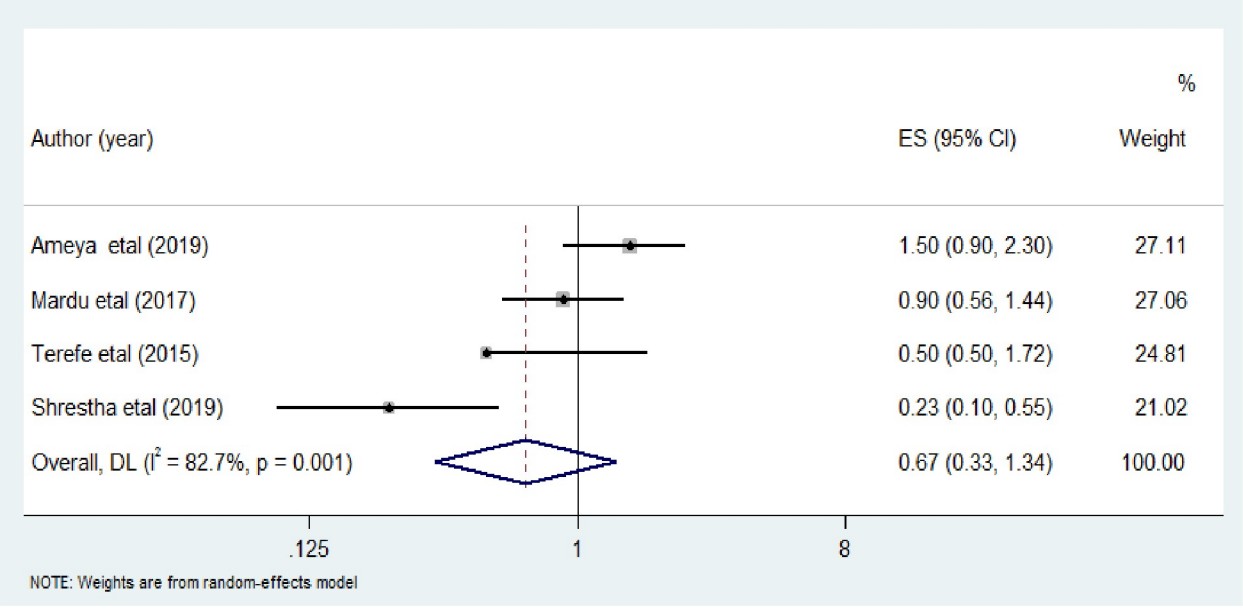

**Fig 7. The pooled odds ratio of the association between residence and intestinal parasitic infection among street dwellers and prison inmates.**

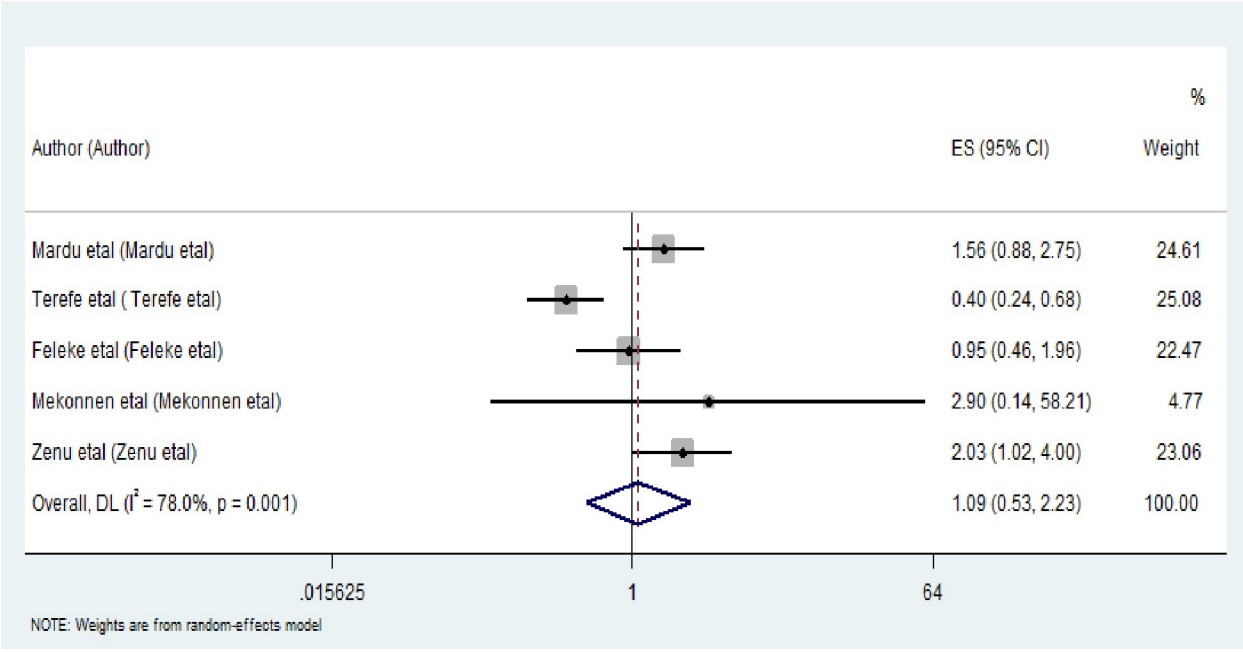

**Fig 8. The pooled odds ratio of the association between finger nail status and intestinal parasitic infection among street dwellers and prison inmates.**

The present study revealed that the pooled prevalence of intestinal parasitic infections among street dwellers and prison inmates was 43.68% (95% CI: 30.56, 56.79). Although there were no studies conducted on the same study groups, the prevalence of intestinal parasitic infections of this study was higher than systematic reviews and meta-analyses conducted among HIV/AIDS patients (39.15%), food handlers working in Ethiopian university cafeterias (28.5%) [38], general populations (25.01%) [39] and food handlers of food service establishments 33.6% [40] in Ethiopia. Similarly, the present study had higher intestinal parasitic infection prevalence compared to studies conducted among pre-school and school children (38%) and general population (34.2%) [41, 42] in Iran. This difference might be due to the fact that street dwellers are socially and economically deprived segment of the population. Hence, lack of clean water for drinking and maintaining personal hygiene, frequent contact with soil and dirty environment and living in unhygienic conditions might increase their exposure for intestinal parasitic infections. Moreover, they might be exposed to intestinal parasitic infections as a result of consumption of contaminated leftover foods. Similarly, prison inmates might also have an increased risk for intestinal parasitic infections due to lack of resource in prison for keeping their personal and environmental hygiene. In addition, they usually living in group and sharing scared resources which might expose them for intestinal parasitic infections due to contamination.

This study also revealed that prevalence of intestinal parasitic infection was lower than other similar studies conducted among school children (46.09%) [43], food handlers at prison in Ethiopia (61.9%) [44] and pre-school and school aged children (48%) in Ethiopia [45]. Similarly, the prevalence of intestinal parasites was lower than a study conducted among children in Karachi, Pakistan (52.8%) [46]. This variation might be due to variation in socioeconomic status, socio cultural beliefs and practices, and geographical location. Moreover,

methodological differences such as study group, sample size, diagnostic method used might contribute for this difference. The current finding is in line with a study conducted in Ethiopia 44.6% [47].

Sub-group analysis was conducted based on study group, sample size, continent of the study conducted, and publication year of the study. The sub-group analysis showed higher intestinal parasitic infections prevalence among street dwellers (68.39%) and studies published from 2000 to 2015 (62.38%). The possible reason might be due to the fact that they are socially and economically deprived. As a result, lack of access for safe water for drinking and maintaining their personal and environmental hygiene. In addition, living in groups and consumption of contaminated foods could expose them for intestinal parasitic infections. The high prevalence of intestinal parasitic infections from 2000–2015 might be due to the fact that the social, economical improvement of the society and the effectiveness of intestinal parasites prevention and control measures implemented worldwide might contribute for the decrease in the burden of intestinal parasitic infections in recent years than before.

With regard to the species of intestinal parasites, *A. lumbricoides* was found to be the dominant intestinal parasite (14.07% (95% CI: 9.15, 18.99)) followed by *E. histolytica* (7.83% (5.47, 10.18)) and *G. lamblia* (7.49% (5.08, 9.90)) in this study. The least common parasite species reported was *E. vermicularis* (1.69 (0.20, 3.18)). Although the magnitude varies these intestinal parasites were the most commonly reported in many studies [40, 42–44, 46, 47]. The prevalence of *A. lumbricoides* (13.98%), Entamoeba spp (16.11%) and *G. lamblia* (9.98%) was higher than the present study [41, 43]. The magnitude of Hook worm and *E.vermicularis* in this study was lower than reports from Ethiopia and Iran [41, 43].

This finding of this study was in line with study conducted in among food handlers at prison in Ethiopia where *A. lumbricoides* (45.6%) is the most dominant parasite, followed by *E. histolytica* (24.1%) [44]. The variation might be due to methodological differences such as the number and group of population studied the special diagnostic tool for species used, the endemicity and weather condition of the study areas.

This study showed that fingernail status had statistically significant association with the prevalence of intestinal parasitic infections AOR: 1.09 (95%CI: 0.53, 2.23). This finding was in line with a meta-analysis study conducted among primary school children in Ethiopia [43, 44, 47].

Although there was no statistically significant association between hand washing habit and intestinal parasitic infection prevalence, the odds of having intestinal parasitic infection occurrence were nearly three times higher in individuals who didn't have hand washing habit(AOR: 2.73, (95% CI: 1.83, 4.07)). In contrast to the finding in the present study, a study conducted among primary school children in Ethiopia [43] showed statistically significant association between hand washing and intestinal parasitic infections. However, it is line with regard to the odds of having intestinal parasitic infections which is increased in those who had hand washing habit.

With regard to the residence, all the studies analyzed were conducted in prison inmates. The statistical test showed that the odds of intestinal parasitic infection among prisoners who were rural resident less likely to be infected by intestinal parasitic infections compared to their counter parts. This was not in agreement with a study conducted among school children [43] in which rural residents are more likely to develop intestinal parasitic infections as compared to those living in urban areas. This might be explained by the fact that in the present study, prisoners were asked their previous resident before detention which might not have any significance as they are living in prison for more than three long ago. Another possible reason might be those prisoners who were living in rural areas before they become prisoner might develop

immunity against intestinal parasites (AOR: 0.67, (95%CI: 0.33, 1.34)) due to the frequent exposure.

## Limitations of the study

Attention to multiple intestinal parasitic infections was not given by many studies. Therefore, the present study could not report the burden of multiple intestinal parasitic infections. Although at least 3 samplings are necessary for standard intestinal parasite diagnosis, all the included studies in this review performed stool samples only once. Furthermore, golden standard methods for the diagnosis of some parasites were not considered which might affect their prevalence. Furthermore, all the included studies were cross-sectional, which might share the limitations of cross-sectional study design.

## Conclusion

In the present study, the overall pooled prevalence of intestinal parasitic infections among street dwellers and prison inmates was relatively high. Fingernail status had statistically significant association with intestinal parasitic infection. The finding of this study should trigger policy makers, governmental and non-governmental organizations and health care providers to give attention for street dwellers and prisoners who are socially deprived segment of the population. The prevention and control of intestinal parasitic infection to reduce the burden of the problem through health education, diagnosis and treatment of infections should also target socially deprived segment of the population such as street dwellers and prison inmates. These groups of the population are usually ignored so there should be population-based studies to more accurately estimate the prevalence of intestinal parasitic infections among street dwellers and prison inmates. Finally, researchers should conduct studies using proper diagnostic methods for each parasite.

## Supporting information

**S1 File. Full electronic search strategy.**
(DOCX)

**S2 File. Results of the individual components of the quality assessment of each study.**
(DOCX)

**S1 Checklist. PRISMA checklist.**
(DOC)

## Author Contributions

**Conceptualization:** Daniel Getacher Feleke.

**Data curation:** Daniel Getacher Feleke, Yonas Alemu, Habtye Bisetegn, Melat Mekonnen, Nebiyou Yemanebrhane.

**Formal analysis:** Daniel Getacher Feleke.

**Methodology:** Daniel Getacher Feleke, Yonas Alemu, Habtye Bisetegn, Melat Mekonnen, Nebiyou Yemanebrhane.

**Project administration:** Daniel Getacher Feleke.

**Software:** Daniel Getacher Feleke, Yonas Alemu, Habtye Bisetegn.

**Supervision:** Daniel Getacher Feleke.

**Visualization:** Habtye Bisetegn, Melat Mekonnen, Nebiyou Yemanebrhane.

**Writing – original draft:** Daniel Getacher Feleke.

**Writing – review & editing:** Daniel Getacher Feleke, Yonas Alemu, Habtye Bisetegn, Melat Mekonnen, Nebiyou Yemanebrhane.

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
