## [Decision Letter · Decision Letter 0]

22 Jun 2021

PONE-D-21-10883

Intestinal parasitic infections and associated factors among street dwellers and prison inmates: A systematic review and meta-analysis

PLOS ONE

Dear Dr. Feleke,

Thank you for submitting your manuscript to PLOS ONE. After careful consideration, we feel that it has merit but does not fully meet PLOS ONE’s publication criteria as it currently stands. Therefore, we invite you to submit a revised version of the manuscript that addresses the points raised during the review process.

More details needed on methodologies and statistics. If the steps of meta-analysis have not been followed, there is need to rework on the data following appropriate methods suggested by the reviewer.

We look forward to receiving your revised manuscript.

Kind regards,

Iddya Karunasagar

Academic Editor

PLOS ONE

Additional Editor Comments:

The reviewers have suggested very important points regarding methodology, which need further clarification and explanation. Please revise considering the reviewer comments.

Journal Requirements:

2. Please confirm that you have included all items recommended in the PRISMA checklist including:

-    the full electronic search strategy used to identify studies with all search terms and limits for at least one database.

-    an explanation for why the search inclusion dates began in 2000

-      the date that the search was last conducted

-    a Supplemental file of the results of the individual components of the quality assessment, not just the overall score, for each study included.

-    See https://journals.plos.org/plosmedicine/article?id=10.1371/journal.pmed.1000100#pmed-1000100-t003 for guidance on reporting.

Thank you.

3. For more information on PLOS ONE's expectations for statistical reporting, please see https://journals.plos.org/plosone/s/submission-guidelines.#loc-statistical-reporting. Please update your Methods and Results sections accordingly.

"We express our gratefulness to Addis Ababa University for all necessary supports."

"The authors received no specific funding for this work.2"

6. We note that this manuscript is a systematic review or meta-analysis; our author guidelines therefore require that you use PRISMA guidance to help improve reporting quality of this type of study. Please upload copies of the completed PRISMA checklist as Supporting Information with a file name “PRISMA checklist”.

Reviewers' comments:

Reviewer's Responses to Questions

**Comments to the Author**

1. Is the manuscript technically sound, and do the data support the conclusions?

Reviewer #1: Partly

Reviewer #2: Partly

2. Has the statistical analysis been performed appropriately and rigorously? 

Reviewer #1: Yes

Reviewer #2: Yes

3. Have the authors made all data underlying the findings in their manuscript fully available?

Reviewer #1: Yes

Reviewer #2: Yes

4. Is the manuscript presented in an intelligible fashion and written in standard English?

Reviewer #1: No

Reviewer #2: Yes

5. Review Comments to the Author

Reviewer #1: The combination of prison inmate and street dwellers lacks rationale as they live in different environment and thus have different risk to parasitic infection. The inclusion of only a handful of papers is not fully justified and does not show the important information and existence of novel information. The finding of nail fingers as risk factor need more clarification as well as other potential factors.

Reviewer #2: I will focus on methods and reporting.

Major

1) Publication bias tests and plots only relevant if you have >10 studies otherwise underpowered to detect much and tend to lead to conclusions that are not justified http://www.ncbi.nlm.nih.gov/pubmed/11106885. If you don’t have enough studies to assess you should discuss this as a major limitation. Even with 10 or 20 studies it is very difficult to visually assess. If you have 20 or more studies it is a considerable strength. Rephrase the methods section to reflect that and also note that publication bias is only relevant in the context of an intervention (i.e. when you look at factors) and not prevalence.

2) The methodological description on the examination of factors is poor and does not provide enough detail (or clearly list the factors). what weighting is used in those models and what models (inverse variance DerSimoniam-Laird, Mantel-Haenszel etc).

3) Meta-analyses of proportions (prevalence) are a bit more complicated since transformations are needed to account for the 0 and 100% limits. Step 1: transformation; step 2: meta-analysis method using standard approach (i.e. inverse variance DerSimonian-Laird); step 3: back-transformation to percentages and plotting. One approach is logit transformation, which is explained in a different context here: http://www.bmj.com/content/352/bmj.i1114. However, a double arcsine transformation is the norm (http://jech.bmj.com/content/early/2013/08/20/jech-2013-203104). The method is implemented in the Stata module metaan http://www.stata-journal.com/article.html?article=st0201. Alternatively you can manually perform using the command you used (I suspect metan, although not referenced). Also see metaprop.

4) Report the confidence intervals for I^2 (calculated using heterogi or metaan in Stata) as argued in http://www.ncbi.nlm.nih.gov/pubmed/17974687. A simple formula exists in the seminal 2002 Higgins paper that proposed I^2.

5) Regarding heterogeneity estimates, all these estimates are very likely off, especially for small meta-analyses, and you should be wary about homogeneity assumptions http://www.ncbi.nlm.nih.gov/pubmed/23922860. So I am not surprised that in the smaller meta-analyses you fail to identify heterogeneity, which very likely exists. Ideally, you would want to check that the findings stand even if you assume high levels of undetected heterogeneity (implemented in Stata metaan) but if not at the very least you need to discuss as a limitation. Personally, I feel that detecting heterogeneity is a good thing since at least it can be incorporated in the model, and I’d pick that any time over a false homogeneity assumption.

Minor

1) Abstract: NOS needs to be defined.

2) Stata not STATA (not an acronym).

3) Cochrane Q is a test

4) Some language corrections are needed (minor).

5) Abstract: the methods need to be described, model used (random effects), what predictors examined, how was heterogeneity assessed etc.

6) don't report results in your methods section. Say, for example, "we decided a priori that if large heterogeneity was observed we would conduct sensitivity analyses".

7) "Furthermore, sensitivity analysis was done to observe the level of heterogeneity" not clear what that means.

8) Year may be worth considering in bias assessment, especially if you don't have enough studies for a formal test: http://www.ncbi.nlm.nih.gov/pubmed/25988604. With newer studies we would be more confident.

9) Reference the commands you used in Stata.

10) How was the random-effect model implemented, i.e. how was heterogeneity estimated? There are numerous ways to do so. Did they use the standard DerSimonian-Laird method? If so, please state so. Also there are better performing methods, for example please see https://www.ncbi.nlm.nih.gov/pubmed/28815652 (or http://www.ncbi.nlm.nih.gov/pubmed/23922860) and the metaan command in Stata where these are implemented (https://www.stata-journal.com/article.html?article=st0201).

11) Did you have to use any continuity corrections or is the outcome not that rare? better to be clear either way.

12) Cochran Q (i.e. chi-square) is notoriously underpowered to detect heterogeneity, especially for small meta-analyses http://www.ncbi.nlm.nih.gov/pubmed/9595615. I would not use

6. PLOS authors have the option to publish the peer review history of their article (what does this mean?). If published, this will include your full peer review and any attached files.

Reviewer #1: No

Reviewer #2: No

---

## [Author Response · Author response to Decision Letter 0]

29 Jun 2021

Dear editor,

We thank the editor and the reviewers for reviewing our manuscript. We have revised the manuscript extensively in the light of the reviewers’ constructive suggestions. A point-by-point response is given below. We have also provided the clean version of the manuscript. All additional information added in the manuscript are marked by red color and yellow highlights. We hope that our revised manuscript will be suitable for publication in PLOS ONE.

Funding: We have made correction on the statement in the acknowledgement section. 

Reviewer #1: The combination of prison inmate and street dwellers lacks rationale as they live in different environment and thus have different risk to parasitic infection. The inclusion of only a handful of papers is not fully justified and does not show the important information and existence of novel information. The finding of nail fingers as risk factor need more clarification as well as other potential factors.

Response: Thank you for your constructive feedbacks. The reason why we decided to study these two groups together because they are the most deprived segment of the population especially in developing countries. Street dwellers are highly affected by low socioeconomic conditions, poor personal and environmental hygiene, and have limited access to clean water. Similarly, prison inmates live in deprived situations with inadequate facilities, malnutrition, scarce potable water, over-crowdedness, and poor hygiene. This is very true especially in developing countries. Besides there were no meta-analysis study on these groups of the population so far. In the present study, in addition to the combined presentation of results, we have also presented the overall pooled prevalence and risk factors for each study group. With regard to the number of studies included, it is true they are few in number. However, we believe they are enough for meta-analysis and can give an insight about the situation of intestinal parasitic infection in these deprived segment of the population. Another proof how these segment of the population are ignored is that the number of researches conducted on these groups are very few. In this meta-analysis we included only factors that was assessed in uniform fashion by the original studies as it is very difficult to Meta-analyze factors that varies in variable category. When we say finger nail status which mean whether the study participants trimmed their finger nail or not. We found statistically significant association between intestinal parasite prevalence finger nail trimming status. Untrimmed finger nail is usually associated with dirt carriage and exposes individuals to parasitic infections.

Reviewer #2: I will focus on methods and reporting.

Response: Thank you very much for rising comments and suggestion that can improve our manuscript in the methods part. 

Major

1) Publication bias tests and plots only relevant if you have >10 studies otherwise underpowered to detect much and tend to lead to conclusions that are not justified http://www.ncbi.nlm.nih.gov/pubmed/11106885. If you don’t have enough studies to assess you should discuss this as a major limitation. Even with 10 or 20 studies it is very difficult to visually assess. If you have 20 or more studies it is a considerable strength. Rephrase the methods section to reflect that and also note that publication bias is only relevant in the context of an intervention (i.e. when you look at factors) and not prevalence.

Response: Thank you very much for your concern. Yes, actually we noted the number of the studies included are just 17. We decided to assess the publication bias because some literatures recommend publication bias assessment if the number of studies are greater than 10 (https://www.ncbi.nlm.nih.gov/pmc/articles/PMC5482177/). Secondly, in this study we were looking for risk factors in addition to the prevalence of intestinal parasitic infection. So, we believe better information can be obtained by assessing publication bias. However, we have removed sentences that explains about publication bias assessments of small studies that included for risk factor assessment. 

2) The methodological description on the examination of factors is poor and does not provide enough detail (or clearly list the factors). What weighting is used in those models and what models (inverse variance DerSimoniam-Laird, Mantel-Haenszel etc).

Response: Thank you. Actually the factors asses in this meta-analysis were, hand washing habit, residence and finger nail status. It was not possible to include other factors from the included studies in meta-analysis because of the variability in factor assessment. The model we used in this study was random effects model. We have mentioned in the result part. We computed this meta- analysis using the DerSimonian-Laired random effects meta-analysis model. We have indicated it in the method section; data processing and analyses sub-section.

3) Meta-analyses of proportions (prevalence) are a bit more complicated since transformations are needed to account for the 0 and 100% limits. Step 1: transformation; step 2: meta-analysis method using standard approach (i.e. inverse variance DerSimonian-Laird); step 3: back-transformation to percentages and plotting. One approach is logit transformation, which is explained in a different context here: http://www.bmj.com/content/352/bmj.i1114. However, a double arcsine transformation is the norm (http://jech.bmj.com/content/early/2013/08/20/jech-2013-203104). The method is implemented in the Stata module metaan http://www.stata-journal.com/article.html?article=st0201. Alternatively you can manually perform using the command you used (I suspect metan, although not referenced). Also see metaprop.

Response: Thank you for your information and resources. We read all the resources you attached and we find them helpful. In this meta-analysis, we used the logit transformation. We have calculated the standard error of the overall intestinal parasite prevalence reported from each study and also standard error of each parasite species prevalence from each study. We have also transformed the odds ratio of the included factors to log odds ratio and we calculated the standard error of logarithm of odds ratio. We used “metan” command for analyses in this meta-analysis and systematic review. 

4) Report the confidence intervals for I^2 (calculated using heterogi or metaan in Stata) as argued in http://www.ncbi.nlm.nih.gov/pubmed/17974687. A simple formula exists in the seminal 2002 Higgins paper that proposed I^2.

Response: Thank you for your suggestion. We have read http://www.ncbi.nlm.nih.gov/pubmed/17974687 and other related articles which made us had a more understanding of heterogeneity and its related values. Although methods and formulas were found in the above mentioned studies, the lack of clarity in assumptions during calculation of 95%CI of I2 and we couldn’t find other studies similar to our study that reported calculated 95% CI of I2 made our effort of calculating 95% CI of I2 unsuccessful. We extraordinarily apologize that it was too difficult to calculate the 95% CI of I2. It seemed there was lack of much defined and easy formulas for calculation of 95%CI of I2, which we finally failed to figure out. Many studies stated that all statistical tests for heterogeneity are weak, including I2. Once again apologize for not addressing your suggestion. 

5) Regarding heterogeneity estimates, all these estimates are very likely off, especially for small meta-analyses, and you should be wary about homogeneity assumptions http://www.ncbi.nlm.nih.gov/pubmed/23922860. So I am not surprised that in the smaller meta-analyses you fail to identify heterogeneity, which very likely exists. Ideally, you would want to check that the findings stand even if you assume high levels of undetected heterogeneity (implemented in Stata metaan) but if not at the very least you need to discuss as a limitation. Personally, I feel that detecting heterogeneity is a good thing since at least it can be incorporated in the model, and I’d pick that any time over a false homogeneity assumption.

 Response: Thank you very much for your support and constructive comment. Yes, we were not able to identify heterogeneity. As you said we also preferred to observe the heterogeneity although we couldn’t identify it rather than assuming homogeneity. So, as per your suggestion we have added a sentence that explain this limitation. 

Minor comments

1) Abstract: NOS needs to be defined.

Response: Thank you we have defined NOS in the abstract section.

2) Stata not STATA (not an acronym).

Response: Thank you for your correction. We did the required correction.

3) Cochrane Q is a test

Response: Thank you. We made the appropriate correction.

4) Some language corrections are needed (minor).

Response: Thank you. We re-read the whole manuscript and made the necessary language corrections.

5) Abstract: the methods need to be described, model used (random effects), what predictors examined, how heterogeneity was assessed etc.

Response: Thank you. We have made corrections on the method part of the abstract by adding some sentences as per your comment.

6) Don’t report results in your methods section. Say, for example, "We decided a priori that if large heterogeneity was observed we would conduct sensitivity analyses".

Response: Thank you for your constructive feedback. We have accepted your comment and removed sentences that states result in the method part. It was just a redundancy as it is already mentioned in the result part. 

7) "Furthermore, sensitivity analysis was done to observe the level of heterogeneity" not clear what that means.

Response: Thank you again for your constructive comment. We are sorry for confusing you. Yes the sentence was not clear. Now, we have corrected it and we hope now it is clear.

8) Year may be worth considering in bias assessment, especially if you don't have enough studies for a formal test: http://www.ncbi.nlm.nih.gov/pubmed/25988604. With newer studies we would be more confident.

Response: Thank you. Yes. Knowing that, we have tried to categorize publication year in to two groups. We have done sub-group analysis to observe the role of publication year in the heterogeneity of the included studies. 

9) Reference the commands you used in Stata.

Response: Thank you. We have mentioned it on the method part and referenced it. 

10) How was the random-effect model implemented, i.e. how was heterogeneity estimated? There are numerous ways to do so. Did they use the standard DerSimonian-Laird method? If so, please state so. Also there are better performing methods, for example please see https://www.ncbi.nlm.nih.gov/pubmed/28815652 (or http://www.ncbi.nlm.nih.gov/pubmed/23922860) and the metaan command in Stata where these are implemented (https://www.stata-journal.com/article.html?article=st0201)

Response: Thank you. We have mention in the result part about type of random effects model we used. We computed computed using the DerSimonian-Laired random effects meta-analysis model. We have indicated it in the method section; data processing and analyses sub-section.

11) Did you have to use any continuity corrections or is the outcome not that rare? Better to be clear either way.

Response: Thank you. We are not clear with this question. If it was about statistical continuity corrections, we didn’t use any. In this study the numbers of studies included were small in number. This shows how researchers and governments are ignored these segment of the population. Although enough studies were not done, the outcome of each study or prevalence was high in both study groups. We will continue our work on this study groups. 

12) Cochran Q (i.e. chi-square) is notoriously underpowered to detect heterogeneity, especially for small meta-analyses http://www.ncbi.nlm.nih.gov/pubmed/9595615. I would not use

Response: Thank you for the advice. We agree with your comment. As you mentioned, Cochran Q (i.e. chi-square) has a low power for estimation of heterogeneity compared to I2. However, it should be highlighted that we used this method just for quantification of heterogeneity.

---

## [Decision Letter · Decision Letter 1]

21 Jul 2021

Intestinal parasitic infections and associated factors among street dwellers and prison inmates: A systematic review and meta-analysis

PONE-D-21-10883R1

Dear Dr. Feleke,

We’re pleased to inform you that your manuscript has been judged scientifically suitable for publication and will be formally accepted for publication once it meets all outstanding technical requirements.

Kind regards,

Iddya Karunasagar

Academic Editor

PLOS ONE

Additional Editor Comments (optional):

All reviewer comments have been addressed.

Reviewers' comments:

Reviewer's Responses to Questions

**Comments to the Author**

1. If the authors have adequately addressed your comments raised in a previous round of review and you feel that this manuscript is now acceptable for publication, you may indicate that here to bypass the “Comments to the Author” section, enter your conflict of interest statement in the “Confidential to Editor” section, and submit your "Accept" recommendation.

Reviewer #2: All comments have been addressed

2. Is the manuscript technically sound, and do the data support the conclusions?

Reviewer #2: Yes

3. Has the statistical analysis been performed appropriately and rigorously? 

Reviewer #2: Yes

4. Have the authors made all data underlying the findings in their manuscript fully available?

Reviewer #2: Yes

5. Is the manuscript presented in an intelligible fashion and written in standard English?

Reviewer #2: Yes

6. Review Comments to the Author

Reviewer #2: I am not ecstatic about the changes made to the paper in response to my previous comments and I don't really understand why the authors struggle to report confidence intervals for I^2 when a formula exists (And I pointed it out to them) and there are user-written commands in Stata that compute it. The fact most studies do not report it does not mean that is good practice.

7. PLOS authors have the option to publish the peer review history of their article (what does this mean?). If published, this will include your full peer review and any attached files.

Reviewer #2: No

---

## [Editor Report · Acceptance letter]

26 Jul 2021

PONE-D-21-10883R1 

Intestinal parasitic infections and associated factors among street dwellers and prison inmates: A systematic review and meta-analysis 

Dear Dr. Feleke:

I'm pleased to inform you that your manuscript has been deemed suitable for publication in PLOS ONE. Congratulations! Your manuscript is now with our production department. 

Kind regards, 

on behalf of

Dr. Iddya Karunasagar 

Academic Editor

PLOS ONE